# Flexible Porous Silicon/Carbon Fiber Anode for High−Performance Lithium−Ion Batteries

**DOI:** 10.3390/ma15093190

**Published:** 2022-04-28

**Authors:** Gang Liu, Xiaoyi Zhu, Xiaohua Li, Dongchen Jia, Dong Li, Zhaoli Ma, Jianjiang Li

**Affiliations:** 1School of Environmental Science and Engineering, Qingdao University, No. 308, Ningxia Road, Qingdao 266071, China; 2019025785@qdu.edu.cn (G.L.); xyzhu@qdu.edu.cn (X.Z.); 2020025847@qdu.edu.cn (D.J.); 2019205603@qdu.edu.cn (D.L.); 2School of Material Science and Engineering, Qingdao University, No. 308, Ningxia Road, Qingdao 266071, China; 2019020442@qdu.edu.cn; 3School of Chemical Experimental Teaching Center, Qingdao University, No. 308, Ningxia Road, Qingdao 266071, China; zlma@qdu.edu.cn

**Keywords:** porous silicon, flexible, binder−free, anode, Li−ion batteries

## Abstract

We demonstrate a cross−linked, 3D conductive network structure, porous silicon@carbon nanofiber (P−Si@CNF) anode by magnesium thermal reduction (MR) and the electrospinning methods. The P−Si thermally reduced from silica (SiO_2_) preserved the monodisperse spheric morphology which can effectively achieve good dispersion in the carbon matrix. The mesoporous structure of P–Si and internal nanopores can effectively relieve the volume expansion to ensure the structure integrity, and its high specific surface area enhances the multi−position electrical contact with the carbon material to improve the conductivity. Additionally, the electrospun CNFs exhibited 3D conductive frameworks that provide pathways for rapid electron/ion diffusion. Through the structural design, key basic scientific problems such as electron/ion transport and the process of lithiation/delithiation can be solved to enhance the cyclic stability. As expected, the P−Si@CNFs showed a high capacity of 907.3 mAh g^−1^ after 100 cycles at a current density of 100 mA g^−1^ and excellent cycling performance, with 625.6 mAh g^−1^ maintained even after 300 cycles. This work develops an alternative approach to solve the key problem of Si nanoparticles’ uneven dispersion in a carbon matrix.

## 1. Introduction

Li−ion batteries (LIBs) are widely used in different fields as a new energy storage technology, including portable electronic devices, electrical transportation, and even energy storage systems, owing to their high energy density, long cycle life, and environmental friendliness [1]. Among all candidate anode materials, silicon (Si) is the most promising anode for the next generation of high−capacity LIBs because of its high capacity (4200 mAh g^−1^), low platform working potential (~0.4 V vs. Li/Li^+^), and rich natural resources [2,3]. However, the volume change (~400%) during the process of lithiation/delithiation [4] and the low electrical conductivity severely limit its commercial development. To mitigate or solve these critical problems, developing nanostructured Si (nanotubes, nanowires, nanoparticles) and Si/C composites represents two effective strategies [5,6]. The combination of high−capacity nanostructured Si with stable conductivity carbon can form a stable SEI and improve the electrode conductivity to obtain a good rate performance and cycling stability [7]. However, nanostructured Si is easily agglomerated, and an incomplete coating of carbon on Si leads to a poor Si–C interface and dispersion and further influences the electrode conductivity and cycling stability [8]. Therefore, achieving a good dispersion of nanostructured Si in a carbon matrix is the key problem in Si/C composite research [9].

Porous silicon (P−Si) nanomaterials have attracted increasing attention owing to their highly porous structure and high specific surface area which are of great significance for enhancing the Li−ion transferring rate and cycling stability [10,11]. Developing porous Si/C composites can exert synergy between the porous structure and carbon matrix [12]. The pores inside the silicon material can relieve the volume expansion and mechanical stress during lithium storage. By dispersing P−Si into the carbon matrix, the overall electronic conductivity of the material can be improved, and the irreversible capacity loss caused by the side reaction can be reduced due to the less direct contact between the porous silicon and electrolyte [13]. Guo et al. [14] fabricated a Si/C hybrid with P−Si nanoparticles loaded in void carbon spheres. The nanoporous structure and special void space are beneficial for the structure integrity and provide channels for the fast transport of electrons/ions during the cycling. The obtained P−Si/C hybrid displayed an improved electrochemical performance. Lu’s team reported a unique yolk–shell structure with a graphene cage encapsulating mesoporous Si spheres. As excepted, the as−synthesized P−Si/C composite displayed extraordinary lithium storage performance in terms of a high specific capacity, a high rate capability, and excellent cycling performance [11].

In recent years, electrostatic spinning technology has been widely used to prepare flexible, binder−free Si/C anode composites [15]. CNFs prepared by electrospinning exhibit 3D carbon frameworks with good mechanical strength, which can greatly accommodate the volume change problem of Si, and can also build a conductive network to further improve the conductivity. Most importantly, this flexible material can be cut into any shape and used as an independent electrode directly without conductive additives, copper foil, and polymer binders, which can greatly improve the energy density [16]. Chen et al. prepared pyrolytic carbon−coated Si/C nanofibers (Si−C/CNFs) via electrospinning, carbonization, and secondary thermal treatment. The Si/C−CNF composite exhibited a more stable cycle performance due to the coating material and nanofiber structure. Si NPs were distributed along the fibers, but the agglomeration was still evident [17]. Therefore, achieving a good dispersion of Si NPs in a carbon matrix is the key problem of the Si/C structure.

Herein, we report a cross−linked, 3D conductive network structure, porous silicon@carbon nanofiber (P−Si@CNF) anode synthesized by magnesium thermal reduction (MR) and the electrospinning method (Figure 1). Firstly, P−Si NPs were prepared through the MR process from SiO_2_ obtained by the well−established Stöber method. P−Si NPs can preserve the original monodisperse spheric morphology of SiO_2_ which is beneficial to the homogeneous dispersion of P−Si NPs in PAN fibers during the subsequent electrospinning process. Meanwhile, the internal nanopores of P−Si can alleviate the volume change, ensuring a uniform stress distribution and the integrity of the structure. Secondly, P−Si NPs were dispersed in PAN to obtain P−Si@PAN fibers via a typical electrospinning process. Thirdly, through a “pre−oxidation−slicing−carbonization” process, the final flexible, binder−free P−Si@CNFs were obtained. In P−Si@CNFs, the higher specific surface area of P−Si enhances the multi−position electrical contact with the carbon material to enhance the overall conductivity. In addition, CNFs, serving as a 3D conductive framework, can not only accommodate the volume expansion of Si but also accelerate the electron/ion transfer. Additionally, P−Si@CNFs can be used as an independent electrode to obtain a high capacity density.

This work develops an alternative approach to improve the uniform dispersion of Si in a carbon matrix which is the key problem of traditional Si/C composites. The porous structure of Si and its high specific surface area allow multi−position electrical contact between Si and the carbon material to effectively realize enhanced electrical conductivity.

## 2. Materials and Methods

### 2.1. Synthesis of SiO_2_ Microspheres

The chemical reagents used in this experiment were obtained from Sinopharm Group Co., Ltd. (Shanghai, China). SiO_2_ microspheres were prepared using the well−established Stöber method [18]. First, 200 mL ethanol–water mixture (V_E_:V_W_ = 4:1) was prepared, and ultrasonic treatment was performed for 3 h to form a uniformly dispersed solution. Then, 5 mL ammonia and 3 mL tetraethylorthosilicate (TEOS) were added separately into the above solution and reacted for 24 h. SiO_2_ microspheres were collected by centrifugation, washed with ethanol–water, and vacuum dried for 12 h. 

### 2.2. Synthesis of P−Si NPs

In the typical MR process, magnesium powders (Mg 99%) and the obtained SiO_2_ microspheres (Mg:SiO_2_ = 1:1) were uniformly mixed well and placed on the end of a graphite boat. Sodium chloride (NaCl: SiO_2_: = 9:1) was placed on the other side. The graphite boat was placed in the center of the quartz tube of the tube furnace (Kejing, Shenzhen, China), heated to 750 °C at a rate of 5 °C min^−1^ in an Ar/H_2_ (5 % H_2_) atmosphere, and maintained for 6 h. After cooling to room temperature, 1 mol L^−1^ hydrochloric acid (HCl) was used to remove NaCl, magnesium oxide (MgO), and other impurities. Finally, the P−Si powders were collected by centrifuging three times with ethanol and vacuum drying for 12 h.

### 2.3. Synthesis of P−Si@CNFs

P−Si@CNFs were synthesized via a typical electrospinning process. Firstly, the P−Si obtained above and 0.5 g polyacrylonitrile (PAN, Mw = 150,000) were dispersed in 4.5 g N, N−DimethylFormamide (DMF) solvent and stirred for 12 h. Then, electrospinning was operated at a voltage of 17.5 kV with a flow rate of 0.75 mL h^−1^. The collected film was named P−Si@PAN. Secondly, the film was heated to 250 °C for 2 h for pre−oxidation at the rate of 2 °C min^−1^ in air. Then, the film was sliced into wafers with a diameter of 1 cm. Finally, the wafers were heated to 850 °C for carbonization for 2 h in an Ar atmosphere. The denoted samples and corresponding synthesis conditions are listed in Appendix A.

## 3. Results and Discussions

The X−ray diffraction (XRD) patterns of SiO_2_ by the well−established Stöber method and the reduced P−Si NPs via the MR process are shown in Appendix A. The broad peak located at about 25° relates to amorphous SiO_2_. After the MR process, three distinct diffraction peaks located at the 2θ values of 28.4, 47.2, and 56.1° were assigned to the crystalline Si phase of the (111), (220), and (311) planes, respectively (JCPDSNO.27−1402) [19,20], suggesting that the spherical amorphous SiO_2_ synthesized by the Stöber method was reduced [21]. The respective XRD patterns of P−Si@CNFs−100, 150, and 200 are shown in Figure 1a. Obviously, besides the typical peaks of Si, the broad peak at 25° related to the (002) plane was observed in the three samples due to the amorphous carbon produced by the pyrolytic carbonization of PAN [22].

The Raman spectra of P−Si@CNFs−100, 150, and 200 are exhibited in Figure 1b. The peak at around 500 cm^−1^ relates to Si. The peak at around 1357 (D band) is due to structure defect− and disorder−induced features in the graphene layers of the carbon, and the peak at 1584 cm^−1^ is attributed to the high−frequency E_2g_ first−order graphitic crystallites of the carbon [23]. The calculated I_D_/I_G_ results of P−Si@CNFs−100, 150, and 200 were all calculated as 0.94. In P−Si@CNFs, the D band is attributed to amorphous carbon derived from the PAN pyrolytic carbonization and corresponds to the broad diffraction peak at 25° in XRD.

To further clarify the proportion of Si content in the composite, thermogravimetric analysis (TGA) was operated under oxygen at the rate of 10 °C min^−1^. Figure 1c shows the TGA curves of P−Si@CNFs−100, 150, and 200. The substantial weight loss between 500 and 700 °C was due to the combustion of amorphous carbon coming from PAN. After 800 °C, the weight increased slightly due to the oxidization of P−Si. Thus, the C content coming from PAN in the P−Si@CNFs−100, P−Si@CNFs−150, and P−Si@CNFs−200 composites was calculated as 74.15, 68.76, and 63.53%, respectively. The corresponding Si content was 24.81, 29.99, and 35.01%, respectively. The TGA curve of the comparative sample Si@CNFs−150 is shown in Appendix A. This sample had about 28.12% Si content, which is similar to the above curves.

The elemental composition of P−Si@CNFs−150 was determined by XPS. Appendix A shows the spectra, confirming the presence of C 1s, O 1s, and Si 2p. Figure 1d reveals the spectrum of Si 2p. It is subdivided into Si−Si including Si2p_1/2_ (98.5 eV) and Si2p_3/2_ (97.6 eV), Si−C (101.5 eV), and Si−O (102.7 eV) [24]. The spectra of C 1s and O 1s are shown in Appendix A. In Appendix A, the spectrum of C 1s is divided into C−Si (283.1 eV), C−C (283.6 eV), and C−O (284.3 eV) [25,26]. In Appendix A, the spectrum of O 1s is divided into O−Si (531.8 eV) and O−C (532.7 eV) [3]. Si−O may come from the residual silica of the MR process and the surface oxidization of the reduced Si [11].

The scanning electron microscopy (SEM) images of the monodisperse spheric SiO_2_ by the Stöber method with a diameter of around 200 nm are shown in Appendix A. The spheric shape was preserved after the MR process (Appendix A). The images of P−Si@CNFs−100, 150, and 200 are displayed in Figure 2. Only few P−Si NPs are distributed in the CNFs in Figure 2a,b. As the content of P−Si increased to 150 mg (Figure 2c,d), P−Si NPs were completely dispersed in the CNFs with little agglomeration. As the P−Si content increased to 200 mg, P−Si NPs presented a distinct agglomeration phenomenon (Figure 2e,f). It can be concluded that P−Si@CNFs−150 with particles uniformly scattered along the CNFs is expected to exhibit good electrochemical performance with the optimum P−Si content. When using commercial Si NPs instead of P−Si, severe particle agglomeration along the CNFs was presented in the comparative sample Si@CNFs−150 (Appendix A). Compared with the commercial Si NPs (80 nm), P−Si with around a 200 nm diameter maintained the original monodisperse spherical morphology, which is beneficial to the uniform dispersion of Si in CNFs.

The transmission electron microscope (TEM) images of SiO_2_ and the corresponding P−Si are displayed in Figure 3a–c. Figure 3b,c reveal that P−Si presented a spheric morphology and was composed of nanosized Si particles with diameters of 5–10 nm. The clearly porous structure can shorten the Li−ion diffusion paths, buffer the Si volume expansion, and facilitate the multi−site contact between the Si and conductive carbon. The Si NPs in Si@CNFs−150 are agglomerated together in Appendix A. Figure 3d shows a typical fiber of P−Si@CNFs−150. It can be found that P−Si NPs were fully embedded and dispersed in the CNFs. The edge of the particles in the CNFs was unsmooth, which confirms the porous structure of P−Si compared with the SiO_2_ nanospheres directly dispersed in the CNFs (Appendix A). In Figure 3e, the high−resolution transmission electron microscopy (HRTEM) image further illustrates that the D−spacing of 0.31 nm was related to Si (111) [27,28,29]. The photo of P−Si@CNFs−150 is shown in Figure 3f. This sample proved to be a flexible electrode due to its good flexible characteristics withstanding multiple bending events.

Figure 4a exhibits the cyclic voltammetry (CV) curve of P−Si@CNFs−150 for the first three cycles. The broad peak at about 0.7~0.85 V in the initial cathodic cycle (lithiation) was attributed to the formation of the irreversible SEI layer and disappeared in the next cycle [30]. The peak at about 0.18 V was due to the lithiation of Si [31]. Correspondingly, the oxidative peak at approximately 0.5 V was due to the dealloying of Li_4_._4_Si [32]. The peaks in the reductive and oxidative cycle became stronger, ascribed to the Li ion diffusion and the electrical conductivity of the electrode. 

Figure 4b shows the initial charge/discharge results of P−Si@CNFs at a current density of 100 mA g^−1^. According to the first discharging and charging capacity process, P−Si@CNFs−100, 150, and 200 displayed an outstanding capacity of 1100.9 and 808.8 mAh g^−1^, 1552.9 and 1241.5 mAh g^−1^, and 1728.7 and 1374.9 mAh g^−1^, with the corresponding initial coulomb efficiencies (ICE) of 73.46, 79.94, and 79.53%, respectively. From the second cycle, the voltage platform between 0.5 and 0.8 V disappeared, which is consistent with the result of the CV curve, confirming that the generated SEI film was stable [33]. The discharge/charge curve of P−Si@CNFs−150 for the first five cycles is shown in Appendix A. The CE reached 95.89% after the second cycle and then increased to 98.02% in the fifth cycle. In addition, the voltage stretch was maintained well in the cycle, which indicates that the electrochemical efficiency was greatly improved. 

The cycling property of P−Si@CNFs−100, 150, and 200 at 100 mA g^−1^ is shown in Figure 4c. After 100 cycles, they displayed a reversible capacity of 680.8, 907.3, and 691.7 mAh g^−1^, respectively. In contrast, P−Si@CNFS−150 displayed a relatively high reversible capacity and better capacity retention owing to the optimal Si content. Si@CNFs−150 delivered poor cycling properties due to the serious aggregation of Si NPs. Appendix A shows the long−term cycle performance of P−Si@CNFS−150 at 100 mA g^−1^, where it maintained a special reversible capacity of 625.6 mAh g^−1^ after 300 cycles, which is consistent with the SEM analysis discussed before. P−Si@CNFS−150 exhibited better dispersion along the CNFs, and the homogeneous mesoporous structure can effectively accommodate Si volume expansion. We disassembled the battery after cycling and show its photo and SEM images in Appendix A. No obvious cracks can be seen, and the electrode microstructure maintained its integrity even after long cycling.

Moreover, the rate performance of P−Si@CNFs−100, 150, and 200 carried out at different current densities is displayed in Figure 4d. Obviously, P−Si@CNFs−150 exhibited the best rate performance. Even after a high current density of 2 A g^−1^, the average reversible charge capacity still reached 420.3 mAh g^−1^. When P−Si@CNFs−150 was cycled at 100 mA g^−1^ again, it still had a high reversible capacity (906.7 mAh g^−1^), confirming the structure stability. P−Si@CNFs−200 displayed a higher initial reversible specific capacity but showed a worse rate performance due to the much higher P−Si content. The excellent rate performance of P−Si@CNFs−150 was attributed to the high specific surface area of P−Si which enhances the multi−site electrical contact with the CNFs, resulting in improved electrical conductivity. Appendix A shows the Brunauer−Emmett−Teller (BET) surface area, pore volume, and average pore size of P−Si@CNFs−150 and Si@CNFs−150. The surface areas of the two samples were 261.67 and 229.35, respectively. Obviously, P−Si@CNFs−150 with multiple pores can facilitate the penetration of the electrolyte into the composite structure and shorten the Li ion diffusion paths. Appendix A shows the schematic diagram of the mechanism of lithiation and delithiation. The P−Si@CNF composite presented a cross−linked 3D conductive framework structure. The original monodisperse spheric morphology realized P−Si uniformly dispersed along the CNFs during the electrospinning process. Meanwhile, the internal nanopores of P−Si effectively alleviated the volume expansion and maintained the structure integrity; thus, the cycling performance was obtained. On the other hand, the Li ion and electron transfer was effectively accelerated owing to the CNF conductive framework and thus improved the electrode rate performance. Additionally, the flexible electrode film obtained by electrospinning can be easily sliced for use as an independent electrode and makes a high capacity density possible. The comparison with other silicon carbon fiber anodes is shown in Appendix A. P−Si@CNFs−150 exhibited a superior capacity and cycling stability, which proves its good structural design.

## 4. Conclusions

In conclusion, a flexible P−Si@CNF electrode with a cross−linked 3D conductive framework microstructure was prepared by the process of magnesium thermal reduction and electrostatic spinning. The optimized P−Si@CNFs with 29.99% Si displayed an excellent capacity (907.3 mAh g^−1^) after 100 cycles at 100 mA g^−1^. Even after 300 cycles, they also maintained a unique capacity of 625.6 mAh g^−1^. The outstanding cyclic stability was attributed to their unique microstructure and flexible characteristics. The reduced P−Si from the MR process exhibited a mesoporous structure which can not only relieve the stress and strain caused by volume expansion but also increase the multi−site electronic contact site between the Si and C, effectively enhancing the overall electrical conductivity. More importantly, P−Si obtained from the reduction of SiO_2_ preserved the single−sphere dispersion in CNFs which is the key factor in structural stability. Overall, the structural design in this work effectively improves the key homogeneous dispersion problem of Si/C composites and has a great prospect in the future application of Si/C anodes in LIBs.

## Data Availability

The data presented in this study are available on request from the corresponding author.

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
