# Peer review of "Flexible Porous Silicon/Carbon Fiber Anode for High−Performance Lithium−Ion Batteries"

_materials, 2022, doi:10.3390/ma15093190_

Round 1

Reviewer 1 Report

The presentation of the submitted manuscript and the proposed approach require some improvements before the publication of the paper. The results are confusing and even need to be verified. 

1)      The novelty of the paper was not introduced clearly. Please explain more about the novelty of the system in a new section.

2)     Introduction is disordered and aimless. Authors just simply list the lows of literature reviews. However, the important information, such as innovations, focused issues, methods, and value of this article, are missing or oversimplified. It is necessary to rewrite the introduction. The literature review has been how written that leads to misunderstanding.

3) Submit a relevant graphical abstract or schematic of the research approach; this may enhance the impact of your paper.

4) The innovation and the importance of this work are not clearly highlighted in the abstract, introduction and conclusions. Please work on this and prove to us why this work is valuable.

5) The manuscript needs a thorough revision of its language and style. Overall, this paper is very difficult to read. Avoid redundancies and keep it short. I suggest a thorough overhaul of the text for a more clear understanding towards the reader.

6) Please add nomenclatures including all variables, parameters, etc to help the reader to follow the paper conveniently.

8) More detail about process modeling and simulation should be given so that other researchers can repeat the results.

9) Compare your results with others from the literature and discuss them. Give more detailed information on the validation procedures.

10) A few more citations to the most recent papers would help clarify the position of this manuscript. to improve the quality, the following recommendations can be incorporated.

"Combining an active method and a passive method in cooling lithium-ion batteries and using the generated heat in heating a residential unit." Journal of Energy Storage 49 (2022): 104181.

"Effect of nano phase change materials on the cooling process of a triangular lithium battery pack." Journal of Energy Storage 51 (2022): 104326.

11) The aim of the conclusions should not be to simply re-state the results: it should explain what they mean, and what are the implications?

12) It is highly recommended that the authors thoroughly check grammar and spelling errors to improve the overall clarity of the paper. Hence, I would recommend for major revision of the manuscript after addressing the significant improvement requirements mentioned above.

Author Response

Dear Reviewer:
Based on your valuable comments, we have made corresponding revisions to the article, and the specific content is in the attachment.

Kind  regards.

Reviewer 2 Report

The manuscript titled "Flexible porous silicon/carbon fiber anode for high performance lithium-ion batteries" is about obtaining self-standing Si anode for Li-ion batteries. At the present, Si/C composite anodes are the most promising candidate to replace conventional graphite anodes, so this topic may be interesting and useful for many readers. Authors used well known synthetic methods (MR and electrospinning) in a new combination. The manuscript is clearly written, experimental design is correct and it is possible to verify the authors’ hypothesis. The authors used a wide range of methods to characterize obtained materials. However, some interpretations are questionable, and authors omitted important experimental details, which makes it difficult to evaluate the results of the work.

There are the following comments for the current manuscript:

1) What is the capacity normalized to? Is it the mass of active materials or total mass of the composite? This is necessary to compare your results with those previously reported. It is customary for energy storage community to normalize capacity per weight of active materials, but for binder-free and self-standing electrodes capacity is more often normalized per total mass.

2) Line 249: «Even after 300 cycles, it also maintains a unique capacity of 625.6 mAh g-1». I think that to confirm the uniqueness of the results it would be useful to add the comparison of the results of this work with the articles that used similar materials. A brief discussion and a table in SI will make the results more vivid.

3) Authors should add more details in experimental part, i.e., type of laser in Raman, type of anode in XPS, model of electrochemical workstation, size and weight of electrodes.

4) Line 146: «The calculated ID/IG results of P-Si@CNFs-100, 150 and 200 are all 0.94, indicating that P-Si@CNFs has high graphitization degree». I suppose that it is a typo, as for high graphitization degree ID/IG should have low value and a sharp peak of 002 graphite planes should be observed in XRD.

5) I recommend adding values of coulombic efficiency to Fig 4c (cycling performance). It is really useful for materials with a notable degree of irreversible process.

6) Line 229: «is attributed to its high specific surface area of P-Si which enhances». This is a reasonable hypothesis, but authors can measure the specific surface area for synthesized and reference materials to support it. One may use gas adsorption method (BET) to do that.

7) There are several problems with the interpretation of XPS spectra. First one is more of a formality, but please use the conventional view of XPS spectra (binding energy decreasing from left to right). Second, the fitting of the HR spectrum of Si 2p is incorrect (Fig 1d). Authors are neglecting spin-orbit splitting for p orbitals and use singlet peaks instead of doublet (2p1/2 and 2p3/2). This must be corrected. Third, the HR spectrum of Si 2p shows that more than half of silicon atoms are in high valence form. Is this surface impurity? In XRD we can only see the peaks for Si and amorphous C. Finally, if you attributed peaks to Si-C and Si-O bond, then good practice supports it by HR spectra of C 1s and O 1s.

8) It is known that self-standing electrodes may have problems with mechanical stability. Have you disassembled the batteries and is there any data on the mechanical stability of the electrodes after cycling?

9) I highly recommend checking the references list and decreasing the number of self-citations, especially in Introduction. Additionally, in line 146 it is stated: «has high graphitization degree [21]» referring to Raman spectroscopy data, yet such method was note used in [21], neither did this reference discuss graphitization degree.

Author Response

(The authors gave the same response as above.)

Reviewer 3 Report

Nice work, but the comments apply to the selected method of obtaining SiO2? ie the Stober method? The surface of unmodified silica particles is mainly coated with silanol groups (−Si − OH), which make it negatively charged. Therefore, the attachment of negatively charged gold nanoparticles thereto is rather limited. the way to solve this problem is to modify silica surface, which leads to a change in the surface charge and, consequently, allows the attachment of gold nanoparticles. the hydroxyl (−OH) groups on the silica surface allow various types of covalent attachment trialkoxyorganosilanes containing functional groups such as: amine (-NH2) or mercaptyl (-SH). After appropriate modification of the silica particles, it is possible to attach many different nanoparticles of materials, including nanoparticles of metals, semiconductors or molecules of biologically active compounds. Where are the parameters for the stober method? or other method being considered. It is known that the method itself significantly influences the physicochemical properties of the obtained silica. Have the authors checked it? Please, expand on this aspect

Author Response

(The authors gave the same response as above.)

Round 2

Reviewer 1 Report

The article can be accepted for publication.

Reviewer 2 Report

Only one a little comment, please add reference list for Fig. S13 in SI. Now the list is only in responses to the reviewer. The manuscript can be accepted in present form.